# Preformation and epigenesis converge to specify primordial germ cell fate in the early *Drosophila* embryo

Megan M. Colonnetta, Yogesh Goyal[¤a¤b], Heath E. Johnson, Sapna Syal[¤c], Paul Schedl, Girish Deshpande*

Department of Molecular Biology, Princeton University, Princeton, New Jersey, United States of America

¤a Current address: Department of Cell and Developmental Biology, Feinberg School of Medicine, Northwestern University, Chicago, Illinois, United States of America
¤b Current Address: Center for Synthetic Biology, Northwestern University, Evanston, Illinois, United States of America
¤c Current Address: Rutgers Robert Wood Johnson Medical School, New Brunswick, New Jersey, United States of America

* gdeshpan@princeton.edu

**Data Availability Statement:** All relevant data are within the manuscript and its Supporting Information files.

## Abstract

A critical step in animal development is the specification of primordial germ cells (PGCs), the precursors of the germline. Two seemingly mutually exclusive mechanisms are implemented across the animal kingdom: epigenesis and preformation. In epigenesis, PGC specification is non-autonomous and depends on extrinsic signaling pathways. The BMP pathway provides the key PGC specification signals in mammals. Preformation is autonomous and mediated by determinants localized within PGCs. In *Drosophila*, a classic example of preformation, constituents of the germ plasm localized at the embryonic posterior are thought to be both necessary and sufficient for proper determination of PGCs. Contrary to this longstanding model, here we show that these localized determinants are insufficient by themselves to direct PGC specification in blastoderm stage embryos. Instead, we find that the BMP signaling pathway is required at multiple steps during the specification process and functions in conjunction with components of the germ plasm to orchestrate PGC fate.

## Author summary

Proper specification of primordial germ cells (PGCs) is crucial as PGCs serve as the precursors of germline stem cells. To specify PGC fate, invertebrates rely upon cell autonomous preformation involving maternally deposited germ plasm. In *Drosophila melanogaster*, to insulate newly formed PGCs from the adverse effects of the cell-cell signaling pathways, germ plasm determinants silence transcription and attenuate the cell cycle. However, our data on the BMP signaling pathway challenge this long-held view of PGC specification and suggest that appropriate specification of embryonic PGCs is sensitive to the BMP ligand, *decapentaplegic* (*dpp*), and its cognate receptor, *thickveins*. We find that PGCs are not only capable of responding to BMP signals from the soma, but also

**Funding:** This work was supported by grants from National Institute of Health (HD093913) to P.S. and G.D., and (GM126975) to P.S. M.M.C. was supported by NSF Graduate Research Fellowship (DGE-1656466). Y.G. was supported by a BWF CASI award. H.E.J. was supported by a Ruth Kirschstein fellowships (F32GM119297). The funders had no role in study design, data collection and analysis, decision to publish, or preparation of the manuscript.

**Competing interests:** The authors have declared that no competing interests exist.

that these signals impact the proper determination of the germ cells. Based on these unanticipated similarities between mammals and flies, we propose a model integrating contribution of both the cell-autonomous (preformation) and non-autonomous (epigenesis) pathways during PGC determination. Consistent with the model, we have observed dominant genetic interactions between, *oskar*, the maternal determinant of PGC fate, and the BMP pathway ligand *dpp*.

## Introduction

Sexual reproduction enables multicellular organisms to transmit genetic information from one generation to the next. The process is initiated by the differentiation of the goniablasts produced by the asymmetric division of male and female germline stem cells (GSCs) and culminates with the fusion of male (sperm) and female (egg) gametes to produce the embryonic zygote. The GSCs of sexually mature animals do not arise *de novo*. Rather, they are derived from a special group of cells, the primordial germ cells (PGCs), that are set aside from the remaining somatic cells during a very early phase of embryonic development. Since the proper specification of PGCs is critical for progression of the reproductive cycle, mechanisms underlying their formation and fate specification have been a major focus of investigation [1–6].

Many characteristic features of PGC specification are broadly conserved in the animal kingdom. One of the key PGC specification steps in most animals is the downregulation of transcription. However, the mechanisms underlying the establishment and/or maintenance of transcriptional quiescence are diverse [6–10]. For instance, in mammals, progenitors of the cells that ultimately give rise to the germline undergo zygotic genome activation (ZGA) like other cells in the embryo; however, after the initial steps in PGC specification, transcription of somatic genes is largely terminated, and these cells begin to revert to an earlier, pluripotent state [6–8]. In worms, the cell lineage that produces the germline is set aside at the first division, and, when transcription commences in the remaining somatic cells at the 3–4 cell stage, it remains transcriptionally quiescent, as do the daughter cells in the lineage destined to become germline [9, 10].

In flies, PGCs are formed after the onset of the minor wave of ZGA, which begins at nuclear cycle (NC) 8. During their formation, ongoing transcription is turned off by *germ-cell less* (*gcl*) [11, 12] while genes activated during the major wave of ZGA (NC14) are kept off by combined action of *polar granule component* (*pgc*) [13–15] and *nanos* (*nos*) [16–18]. Broad downregulation of transcription is reflected in the phosphorylation status of the RNA polymerase II CTD domain. PhosphoSer2, a modification correlated with transcriptional elongation is largely absent in newly formed PGCs, while PhosphoSer5, a modification linked to initiation, is substantially reduced [14, 17, 19, 20]. A second common feature is alterations in histone modification profiles. In worms and flies, histone H3meK4, a modification associated with active transcription, is largely absent in young PGCs. In both organisms, the germline determinant *nos* contributes to the inhibition of this histone modification [21]. Besides *nos*, suppression of the H3meK4 modification in flies requires the H3meK4 demethylase, *Su(var)3-3*, the Swi/Snf chromatin remodeling complex subunit, *osa* [14], and the transcriptional regulator *pgc* [13, 14]. There are also alterations in the heterochromatic histone modification, H3meK9 [21]. Like worms and flies, mouse PGCs also display changes in histone modifications [20, 22, 23]. A third common feature is pausing the cell cycle. In worms, flies, and mammals, PGCs arrest cell cycle in G2 [17, 20, 24–26]. In addition to these common characteristics, genes implicated

in PGC specification in model invertebrates (flies or worms), like *nos*, *vasa*, and *piwi*, are also conserved in higher animals (mice, human etc.) [1, 8, 9].

While many of the characteristics of PGCs that distinguish them from the soma are widely shared amongst different animal species, there is one striking dichotomy, namely whether the mechanism driving specification is "epigenesis" or "preformation." In epigenesis, specification is non-autonomous and depends upon cell-cell signaling. In preformation, specification is autonomous and is driven by determinants that are localized in the presumptive PGCs. Mammals utilize epigenesis. In pre-implantation embryos, a combination of inductive signals from extraembryonic ectoderm and visceral endoderm acts to induce cells within the posterior epiblast to become PGCs [6, 7, 27]. Signaling appears to be at least a two-step process in which Wnt3 (Wingless 3) first primes cells in the epiblast. Once primed, the cells can respond to the BMP (bone morphogenetic protein) ligands Bmp4 and Bmp8b, which are secreted by the extraembryonic ectoderm, and Bmp2, which is secreted by the visceral endoderm [28–31]. These signals activate Smad transcription factors that set the transcriptional program required for PGC specification in motion [32–34]. By contrast, worms and flies are thought to employ an exclusively preformation mechanism involving localized cell-autonomous factors. In worms, localized PGC determinants segregate to only one of the daughter cells during each of the blastomere cell divisions that ultimately give rise to the two PGC founders, Z2 and Z3 [9, 35]. In fly embryos, fertilization is followed by a series of rapid synchronous nuclear divisions which culminate in cellularization at the end of NC14 [36]. However, earlier, during NC9, several nuclei migrate to the posterior pole of the embryo and induce the formation of pole buds [37]. During bud formation, the centrosomes/microtubule network associated with each incoming nucleus triggers the release of localized PGC determinants from the posterior cortical cytoskeleton, and these factors are then incorporated into the newly formed PGCs during cellularization [37, 38]. When these factors are not properly sequestered in the newly formed PGCs, PGC specification fails [12, 38–40].

The master determinant that orchestrates PGC formation and subsequent specification in flies is *oskar* (*osk*) [41]. *osk* mRNAs are localized to the posterior pole of the oocyte during oogenesis and translated during mid-oogenesis [42]. Osk protein then mediates the recruitment and assembly of key components of the germ plasm including Vasa, Tudor, Valois, and Aubergine proteins and *pgc*, *gcl*, and *nos* mRNAs. Once assembled at the posterior of the egg, the germ plasm is sufficient to induce the formation of PGCs. Illmensee and Mahowald showed that injection of pole plasm at the anterior of the embryo induces the formation of ectopic PGCs [43]. This finding was recapitulated by Ephrussi and Lehmann, who replaced the *osk* 3' UTR with a *bicoid* (*bic*) 3' UTR [44]. They found that pole plasm assembled by the ectopic Osk protein was sufficient to induce the formation of fully functional PGCs at the anterior of the embryo. Furthermore, overexpression of *osk* increases the number of PGCs and can also induce ectopic PGCs on the dorsal side of the embryo [45].

These and other findings have reinforced the idea that a preformation mechanism—relying exclusively on localized determinants—is responsible for PGC specification in flies. However, later in embryogenesis, PGCs are neither indifferent nor immune to extracellular signals. As is the case in other organisms, fly PGCs must migrate from their site of formation at the posterior pole through the embryo to coalesce with the somatic gonad precursor cells (SGPs) [46]. Their migration through the mesoderm is mediated by the Hedgehog (Hh) signaling pathway, and they are directed by potentiated Hh ligand produced by the SGPs [47–49]. During roughly the same time frame (stage 10–14 of embryogenesis), fly PGCs not only respond to BMP signaling but are dependent on BMP signals to maintain their identity. When the BMP receptor *thickveins* (*tkv*) is knocked down using the germline-specific *nos-Gal4* driver, Vasa protein accumulation in the PGCs is disrupted while the assembly of the germline-specific spectrosome is

compromised [50]. While *hh* directed PGC migration appears to be mediated by a novel non-transcriptional pathway [49, 51], maintenance of PGC fate in stage 9–14 embryos depends upon canonical transcription factors downstream of BMP. *Nos-Gal4* dependent RNAi knockdown of the Smad co-factor *medea* disrupts Vasa protein accumulation. Similarly, overexpression of *smurf*, a ubiquitin E3 ligase that targets phosphorylated (and active) Smad for degradation, induces defects in Vasa accumulation and spectrosome assembly [50].

These findings show that during mid-embryogenesis, in the period leading up to gonad coalescence, the BMP pathway not only helps maintain PGC identity but also functions to promote PGC differentiation into GSCs. This would suggest that, at least in this time frame, maternal determinants alone are not sufficient for either maintenance of PGC identity or promoting differentiation. If this is the case, a relevant question is whether localized maternal determinants are on their own sufficient for the specification of PGCs in pre-cellular blastoderm embryos or if they also require input(s) from the BMP signaling pathway even at this stage of development. Here we have addressed this question.

## Results

### Early embryonic overexpression of *dpp* results in a modest increase in PGC number due to enhanced mitosis

Studies by Dorfman and Shilo showed that newly formed PGCs in syncytial and cellular blastoderm embryos are not immune to the BMP signaling pathway [52]. Like their somatic counterparts on the dorsal side of the embryo, activated pMad accumulates to high levels in PGC nuclei, and this accretion depends on both the *tkv* receptor and its ligand, decapentaplegic (*dpp*) [53–55]. As would be expected from the accumulation of pMad in PGC nuclei, previous studies on the expression of *dpp* mRNA and the Dpp ligand have shown that though the *dpp* expression domain in early embryos is restricted to the dorsal side of the embryo, it encompasses the entire posterior (and anterior) pole of the embryo [56]. In germline clone embryos homozygous for a *tkv* mutation (*FRTtkv$^8$* referred to as *tkv$^{m-}$* henceforth), pMad is completely lost in the soma and in newly formed PGCs. When *tkv$^{m-}$* embryos are fertilized by a WT *tkv* sperm, zygotic expression can partially mitigate defects in pMad accumulation both in the soma and in newly formed PGCs. This would indicate that unlike many other genes whose transcription is specifically turned off in PGCs, transcription of *tkv* is not [52]. The fact that *tkv* is amongst the few genes known to be transcribed in newly formed PGCs suggests that pMad accumulation may have an important function. On the other hand, it is also possible that pMad-dependent induction of downstream targets is blocked by the maternal determinant(s) that downregulate global transcription in newly formed PGCs. In this case, the PGCs would be immune to the normal activity of nuclear pMad.

To test whether newly formed PGCs are responsive to inputs from the BMP signaling pathway, we generated an excessive level of Dpp using *twist-Gal4* to drive expression of a *UAS-dpp* transgene on the ventral side of the embryo. Consistent with the idea that PGCs retain the ability to respond to BMP signaling, likely via the phosphorylation and nuclear localization of pMad, we find that excess Dpp results in a modest but significant increase in the number of PGCs both in stage 4 syncytial blastoderm (S1A and S1B Fig and S1 Table) and in stage 5/6 i.e. late syncytial/cellular blastoderm stage embryos (S1C and S1D Fig) (26.4 PGCs/ embryo in *twi-Gal4*/*UAS-dpp*, n = 16; versus 21.9 PGCs per embryo, n = 20 in control; p = 0.0019 by t-test). A similar elevation in PGC count was also observed in syncytial /cellular blastoderm stage embryos when Dpp levels were enhanced using maternal *tubulin-Gal4* (S1 Table; p = 0.027 by t-test).

Since WT PGCs divide only 0–2 times after they are formed and then cease division by the time the embryo cellularizes [57], the increase in the number of PGCs could be due to a failure to fully exit the cell cycle. If this idea is correct, the frequency of PGCs in mitosis should be elevated in *twi-Gal4/UAS-dpp* embryos. To test this possibility, we identified PGCs in mitosis using phospho-Histone 3 (pH3) antibody [58, 59]. In WT, pH3 is detected relatively infrequently in PGCs in the period between their formation and the end of NC14 (Fig 1A and S2 Table; 0.7 pole cells per embryo, n = 18). By contrast, this number is considerably elevated (~4 fold) in *twi-Gal4/UAS-dpp* embryos as shown in Fig 1B (2.8 pole cells per embryo, n = 18; p = 1.6e-5 by t-test) (Fig 1B and S2 Table). This suggests that the Dpp signaling pathway promotes proliferation of newly formed PGCs.

To further analyze how excess Dpp might promote PGC division, we examined expression of Cyclin B (CycB). In WT PGCs, translation of *cyclin B* mRNA is specifically targeted for repression by the Nanos:Pumilio complex [25, 26]. Therefore, only very low levels of CycB are detected in WT PGCs (Fig 1C). In contrast, in *twi-Gal4/UAS-dpp* embryos, CycB is readily detected in many PGCs (Fig 1D and S2 Table) (0.3 PGCs in control, n = 9, versus 2.2 PGCs in *twi-Gal4/UAS-dpp* embryos, n = 9; p = 0.0045 by t-test). The increase in CycB levels in *twi-Gal4/UAS-dpp* embryos indicates that Nos-dependent inhibition of *cyclin B* translation is disrupted. Interestingly however, we did not observe any obvious reduction in Vasa levels (S2 Fig and S3 Table; p = 1.0 by Fisher's exact test) in *twi-Gal4/UAS-dpp* PGCs.

## PGCs maintain transcriptional quiescence in the presence of excess BMP signaling

Accompanying the shutdown of mRNA transcription in newly formed PGCs, the phosphorylation of Ser2 and Ser5 in the heptad repeats in the CTD (C-terminal domain) of the large Pol

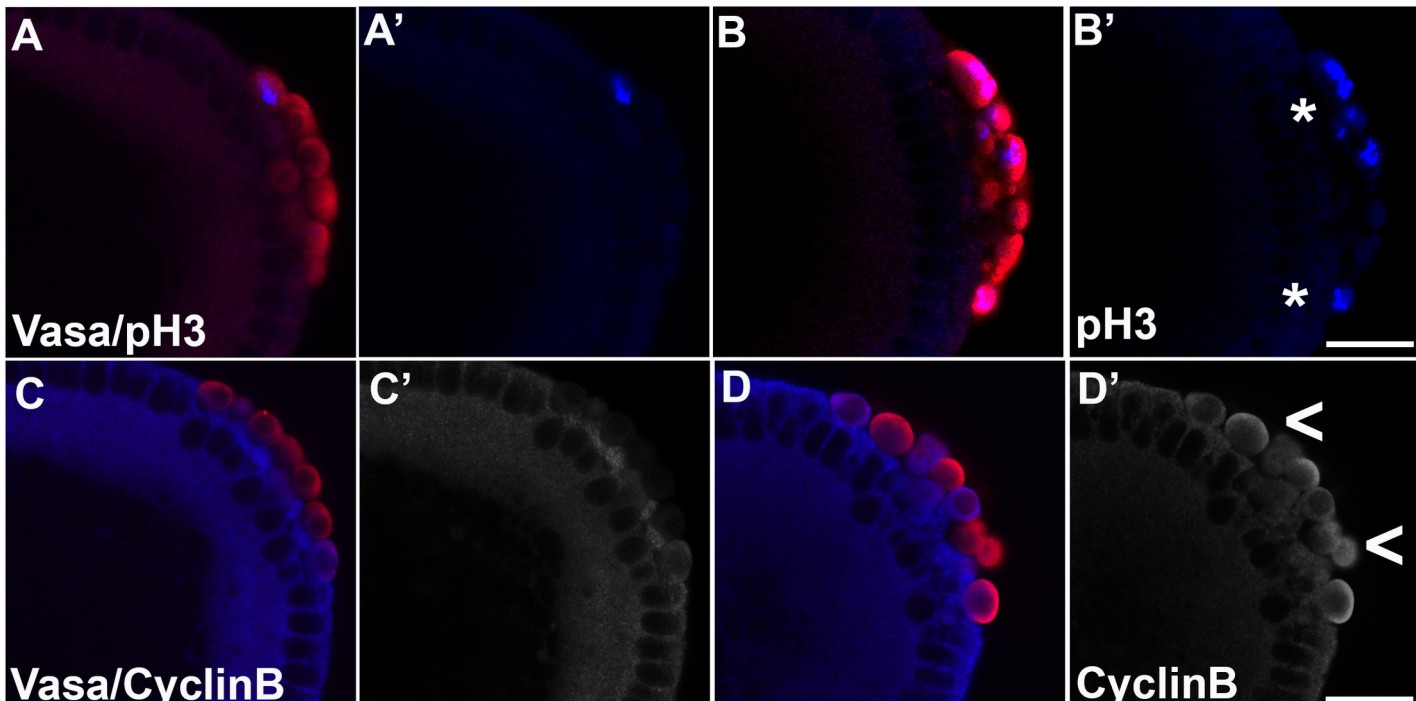

**Fig 1. Ectopic expression of *dpp* using *twist-Gal4* enhances mitosis, resulting in a modest increase in PGC count.** (A/C) *twi-Gal4* (B/D) or *twi-Gal4/UAS-dpp* embryos were stained for PGC marker Vasa (A-D, red) and either pH3 (A'-B') or Cyclin B (C'-D') (blue) to evaluate mitotic state of pole cells. Excess mitotic marker compared to WT levels in PGCs is indicated by asterisks (pH3) or carets (Cyclin B). Scale bar represents 10 μm.

II subunit is substantially reduced [13, 14, 19, 21, 60]. However, after transcription is upregulated in PGCs, the level of both CTD modifications increases [19]. Since phosphorylation and nuclear localization of pMad modulates transcription in somatic cells [61], we wondered whether excess Dpp disrupts the global downregulation of transcription normally observed in WT PGCs. As the accumulation of pSer2 faithfully reports on transcriptional elongation, we examined the levels of pSer2 in the *twi-Gal4/UAS-dpp* and WT PGCs. However, there was no detectable difference between *twi-Gal4/UAS-dpp* and WT embryos (S2 Fig and S3 Table; p = 0.676744 by Fisher's exact test). Thus, PolII transcription in PGCs is not detectably increased by excess Dpp.

## Vasa accumulation in newly formed PGCs depends upon the BMP signaling pathway

The increase in mitotic activity induced by excess Dpp indicates that PGCs are neither indifferent nor immune to BMP signaling. A plausible inference from this finding is that, as is the case in mammals, BMP signaling contributes to the specification of PGC identity in blastoderm stage fly embryos. To start exploring this possibility, we first examined Vasa protein accumulation in PGCs of two different genetic backgrounds. The first is a viable, partial loss of function (LOF) allele of the *dpp* ligand, *dpp^{hr92}* [62], while the second is a viable partial LOF allele of the *tkv* receptor, *tkv^{427}* [63]. pMad levels have been shown to decrease in the embryos compromised for components of *dpp* signaling, including *dpp* itself [52]. Accompanying this reduction in pMad, there is a decrease in Vasa, a marker of germline identity (Fig 2B). For *dpp^{hr92}*, we found that 38% of the *dpp^{hr92}* PGCs had noticeably reduced levels of Vasa (n = 63) compared to only 8% for WT (n = 49) (S4 Table, p = 0.00325 by Fisher's exact test). In the case of *tkv^{427}* embryos, 32% had reduced levels of Vasa (n = 53), while Vasa was reduced in only 7% of WT syncytial blastoderm embryos (Fig 2C and S4 Table; p = 0.00304 by Fisher's exact test). A similar result was obtained when we analyzed Vasa in PGCs of embryos produced by *tkv^{m-}* germline clone mothers mated to WT fathers (*tkv*^{m-z+}). We found 32.8% of *tkv*^{m-z+} PGCs (n = 48; p = 3.4e-5 by Fisher's exact test) showed reduced levels of Vasa compared to 2.1% of PGCs in the control syncytial blastoderm embryos (n = 47, S4 Table).

These results show that BMP pathway components impact accumulation of Vasa, a PGC-specific marker, early in embryogenesis. While this observation indicates that BMP signaling impacts PGC specification, there is a potential complication. In all three experiments, either the mother or the maternal germline was homozygous for a mutation in *dpp* or *tkv*. While

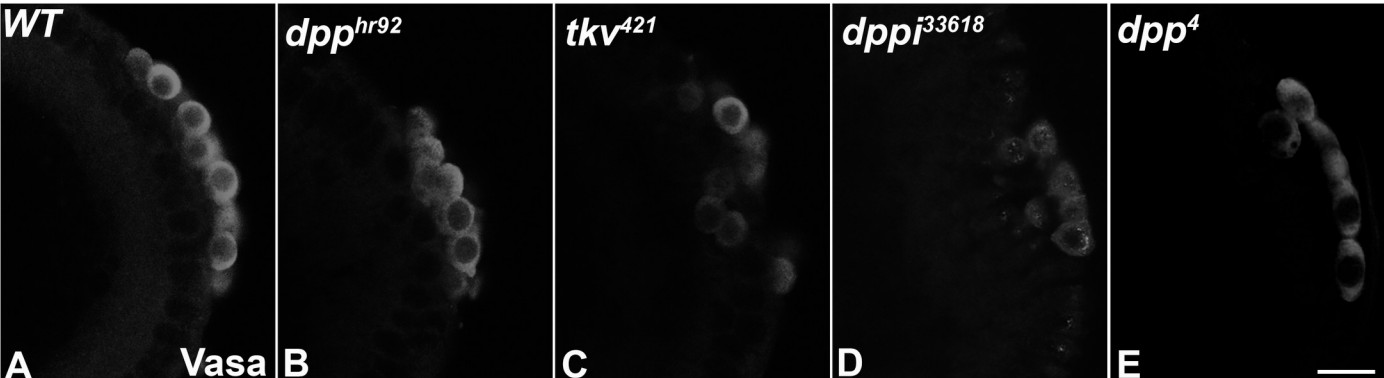

**Fig 2. Embryos compromised for BMP signaling exhibit loss of Vasa.** Embryos of indicated genotype were stained for the pole cell marker Vasa (white) to visualize PGC number and Vasa levels. (A) *WT* (B) *dpp^{hr92}* (C) *tkv^{427}* (D) *dppi^{33618}* (E) *dpp^4*. Scale bar represents 10 μm.

there is no known role for the BMP pathway in the assembly of functional pole plasm during oogenesis, this remains a formal possibility. For this reason, we took two different approaches to ascertain whether the effects on Vasa are strictly zygotic, rather than a consequence of some undocumented role for BMP signaling in pole plasm assembly/functioning. In the first, we used two *UAS:dppRNAi* transgenes (*dppi*[33618] and *dppi*[25782]) to knock down expression of *dpp* in blastoderm stage embryos. In these experiments, we mated mothers carrying a *mat-tubulin-Gal4* transgene to fathers homozygous for each *UAS-dpp RNAi* transgene. Knockdown of *dpp* in pre-cellular blastoderm embryos disrupts Vasa accumulation in newly formed PGCs (Fig 2D). We found that 31.7% of the PGCs (n = 145) had reduced Vasa compared to 2.9% of the PGCs (n = 136) in control embryos in which *mat-tublin-Gal4* mothers were mated to fathers carrying an *UAS:egfpi* transgene (S4 Table; p = 0.0 by Fisher's exact test). Though somewhat less effective, the *dppi*[25782] transgene gave similar results (21.1% of the PGCs, n = 180) had reduced levels of Vasa (S4 Table; p = 1.0e-6 by Fisher's exact test).

In the second approach, we examined Vasa accumulation in PGCs of embryos homozygous for a strong loss-of-function mutation of *dpp*, *dpp*[4]. In order to identify homozygous *dpp*[4] blastoderm stage embryos, the *dpp* mutation was recombined with a *twist* protein null. Embryos collected from a cross between heterozygous *dpp*[4]*twi/CyO* parents were probed with a combination of Vasa and Twi antibodies. As was observed in the zygotic *RNAi* knockdowns, Vasa levels were reduced in PGCs of homozygous *dpp*[4]*twi* embryos. We found that 27.8% of the PGCs (n = 248) in embryos that had no detectable Twi protein had reduced levels of Vasa, compared to 0.9% of the PGCs (n = 216) in the Twi-positive siblings (S4 Table; p = 0.0 by Fisher's exact test).

## Transcriptional quiescence is partially compromised when the BMP pathway is disrupted

The loss of Vasa in newly formed PGCs suggests that the BMP pathway functions at an early step in PGC specification. If so, other hallmarks of newly formed PGCs might also be disrupted. One of these is the establishment of transcriptional quiescence. To assess the impact of the BMP pathway on downregulating RNA Pol II transcription, we used several approaches. In WT, the signature for transcriptional elongation (pSer2) is absent in syncytial blastoderm stage embryo PGCs while it is readily detected in somatic nuclei. As shown for two different embryos (Fig 3B and 3C), newly formed PGCs (marked by Vasa protein) in *dpp*[hr92] embryos accumulate levels of pSer2 approaching that of surrounding somatic nuclei. Quantification of pSer2-positive PGCs indicates that 23.9% of the syncytial blastoderm stage *dpp*[hr92] PGCs (n = 67; p = 0.000597 by Fisher's exact test) and 30% (n = 120; p = 1.0e-6 by Fisher's exact test) of *tkv*[427] PGCs have nuclear pSer2, as compared to 5.1% for WT PGCs (n = 98) (Figs 3 and S3 and S5 Table).

To further test the effects of the BMP pathway on RNA polymerase II activity, we examined the PGCs of embryos laid by *tkv* germline clone mothers (*tkv*[m-z+]). While PGCs in the WT control (Fig 3) only infrequently had detectable levels of pSer2 (4.6%, n = 43), over forty percent (41.2%, n = 51) of PGCs in the *tkv*[m-z+] blastoderm stage embryos had pSer2 (S5 Table; p = 2.70e-5 by Fisher's exact test). Moreover, those *tkv*[m-z+] PGCs with the highest levels of pSer2 often showed evidence of Vasa depletion (carets, Fig 3B and 3C).

pSer2 is not the only marker of transcriptional activity that is altered in PGCs when components of the BMP pathway are not fully active. In WT, the chromatin marker of active transcription, the histone H3 modification, H3meK4, is upregulated in the somatic nuclei of syncytial blastoderm embryos, while it is almost completely absent in PGC nuclei. In contrast, this marker is readily detected in *tkv*[427] PGCs (51.7% of *tkv*[427] PGCs, n = 56; p = 0.0 by Fisher's

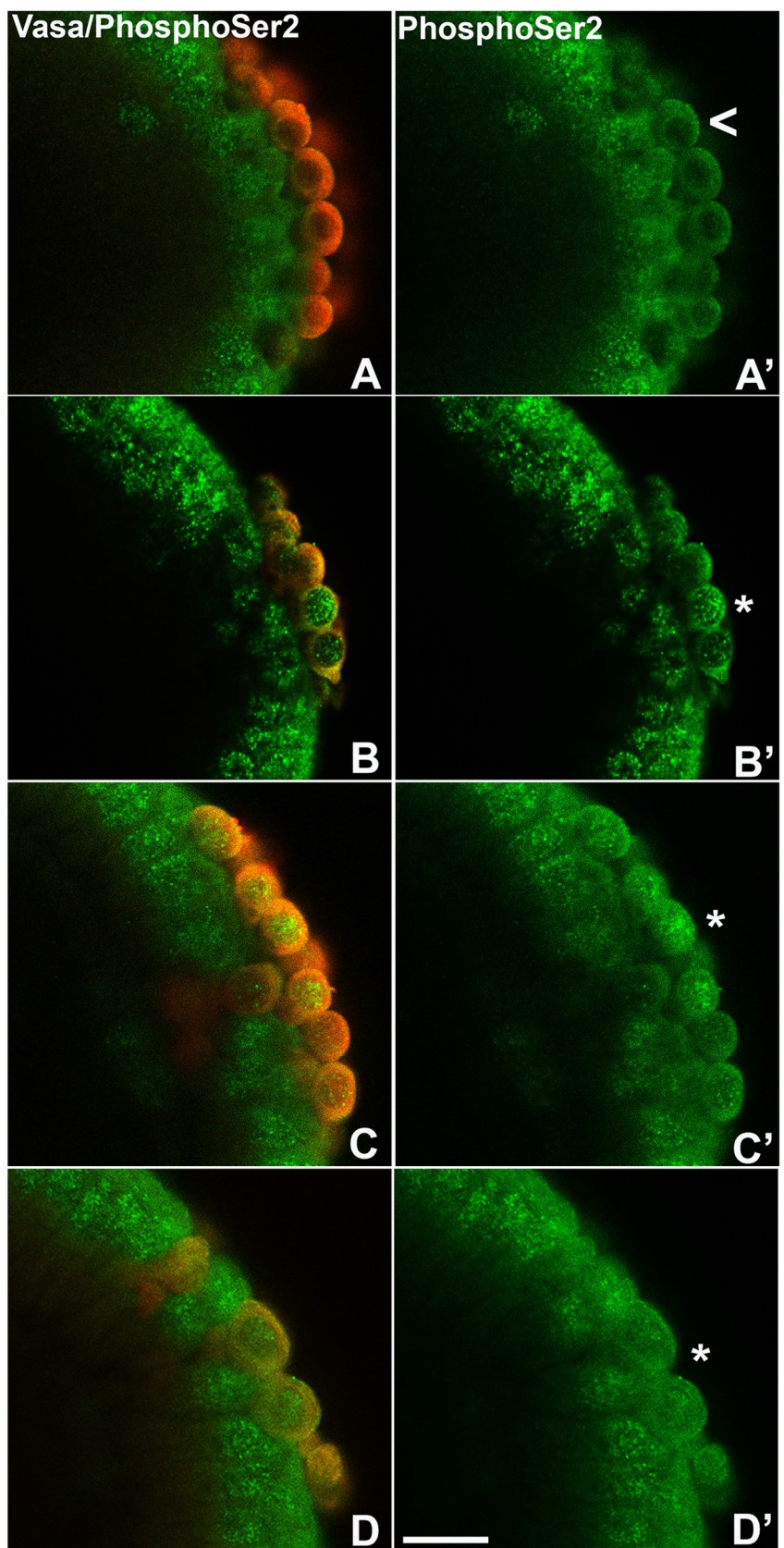

**Fig 3. Loss of BMP signaling components leads to elevated levels of a marker of active transcription in PGCs.** (A) *WT*, (B) *dpp^{hr92}*, (C) *tkv^{427}*, and (D) *tkv^{m-z+}* embryos were stained for pole cell marker Vasa (red) and phosphoSer2 (an indicator of transcriptional activation, green). Asterisks indicate presence of phosphoSer2 in PGCs. Scale bar represents 10 μm.

exact test), as opposed to 2.9% of WT PGCs (n = 69) (S4 Fig). Furthermore, weaker but significant elevation in the H3meK4 signal was also observed in the PGCs from *dpp^{hr92}* embryos (22.2%, n = 54; p = 0.001077 by Fisher's exact test) (S5 Table). Taken together, these observations suggest that loss of BMP signaling in early embryos disrupts the proper downregulation of transcription in PGCs.

## Genes normally silenced in newly formed PGCs are ectopically expressed when the BMP pathway is compromised

Previous studies have shown that the three maternal factors known to be responsible for repressing transcription in newly formed PGCs, Gcl, Nos, and Pgc, function at different times and target an overlapping set of genes. *gcl* functions during PGC cellularization. Gcl is responsible for shutting off transcription of genes that are activated in the minor wave of ZGA. Its targets include *scute* (*sis-b*), *sis-a*, and *runt* [11, 12]. It also has a role in silencing the *Sxl* establishment promoter, *Sxl-Pe*. Pgc and Nos function after PGC formation but have different activities and gene targets. *pgc* inhibits the kinase, pTFb, that phosphorylates the Ser2 residue in the Pol II CTD domain. Its known targets include *tailless* (*tll*) and *slow as molasses* (*slam)* [13, 14]. While it is not understood how *nos* inhibits Pol II transcription, its known targets include *fushi-tarzu*, *even-skipped*, and the *Sxl-Pe* promoter [17].

To test the effects of compromising BMP signaling on transcriptional activity, we selected *tll*, *slam*, and *Sxl-Pe* and examined their expression using a combination of fluorescent *in situ* hybridization (FISH) and single molecule FISH (smFISH). The *tll* and *slam* probes were directed against sequences in the corresponding mRNAs while the *Sxl-Pe* probe is homologous to a large intron in the *Sxl-Pe* transcript and thus only detects nascent mRNAs. As shown in Fig 4, we find that *tll* mRNA is expressed in a subset of *dpp^{hr92}* PGCs (Fig 4). While all WT embryos had no *tll* transcription in PGCs (n = 19), 70.4% of *dpp^{hr92}* embryos displayed ectopic *tll* transcription (n = 27; p = 1.0e-6 by Fisher's exact test). To further quantify *tll* mRNA expression in *dpp^{hr92}* PGCs compared to WT, we normalized *tll* staining intensity in PGCs with respect to adjacent somatic cells. As shown in the plot, there is a significant increase in the average level of *tll* signal in the *dpp^{hr92}* PGCs compared to WT (Fig 4H). In WT embryos, *slam* transcripts begin to appear at syncytial blastoderm stage, and the levels rise considerably in cellular blastoderm stage embryos. *slam* RNA also associates with Slam protein, which decorates membranes [64]. Consequently, *slam* probe directed against the mRNAs labels the extending membrane compartment of somatic nuclei/cells. *slam* is not, however, transcribed in PGCs and they are normally devoid of *slam* mRNA. Using smFISH, we detected *slam* expression in a subset of the *dpp^{hr92}* PGCs (Fig 5B: see asterisks). Approximately 26.3% of the *dpp^{hr92}* (n = 38; p = 0.000991 by Fisher's exact test) embryos have one or more PGCs that express *slam* mRNA (S6 Table). We also examined PGCs in embryos carrying the *dpp* RNAi line *dppi^{33618}* (Fig 5C: see asterisks). We find that 35.7% of the *dppi^{33618}* express *slam* (n = 14; p = 0.000852 by Fisher's exact test) as compared to 0% for WT embryos (n = 37). For *Sxl-Pe*, the frequency of *dpp^{hr92}* embryos that that have detectable transcripts is lower that for *tll* and *slam*. We find that 18.2% of the *dpp^{hr92}* embryos (n = 11; p = 0.47619 by Fisher's exact test) have *Sxl-Pe* transcripts, while *Sxl-Pe* transcripts are not observed in WT embryos (n = 10) (S6 Table). It seems likely that we detect *Sxl-Pe* transcripts much less frequently in *dpp^{hr92}* PGCs because the *Sxl-Pe* probe only

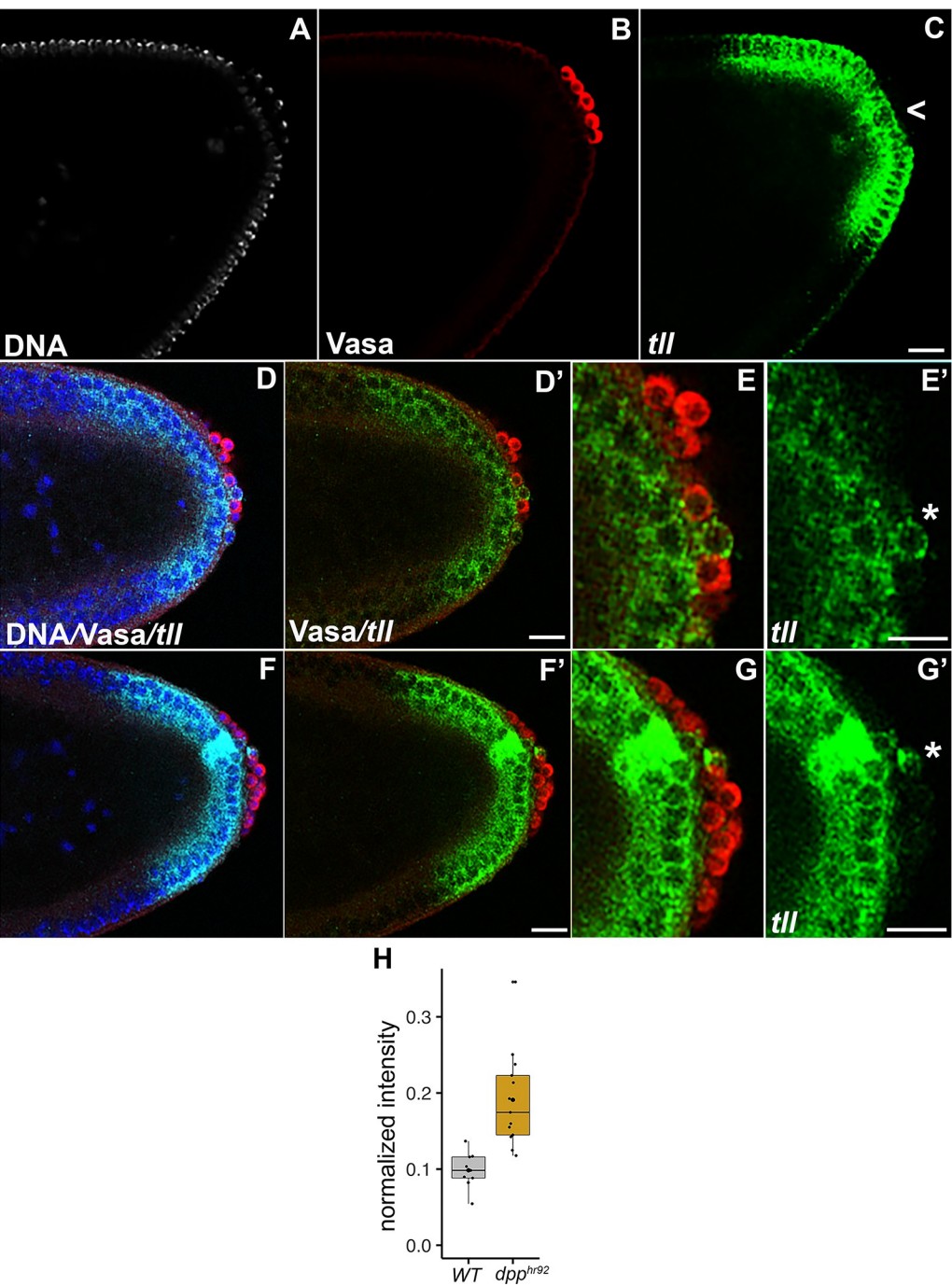

**Fig 4. *dpp*-compromised embryos show PGC-specific aberrant transcription of the terminal patterning gene *tailless* (*tll*).** (A-C) *WT* and (D-G) *dpp^{hr92}* embryos were simultaneously stained for Vasa (red) and probed for *tll* RNA (green) using FISH. Nuclei were labeled using Hoescht (white/blue). Panels E and G show enlarged regions of Panels D' and F', respectively. Scale bar represents 10 μm. H) Plot below displays normalized density value of *tll* transcripts in PGCs relative to soma in the same embryo (see Materials and Methods for details of quantification).

hybridizes to nascent transcripts, and transcription bursts are relatively infrequent. Furthermore, we have found that ectopic *Sxl-Pe* expression does show a slight sex bias in *gcl* embryos [12]. A similar bias may render detection of *Sxl* transcripts relatively rarer compared to the sex-nonspecific somatic genes (e.g. *slam*, *tll*).

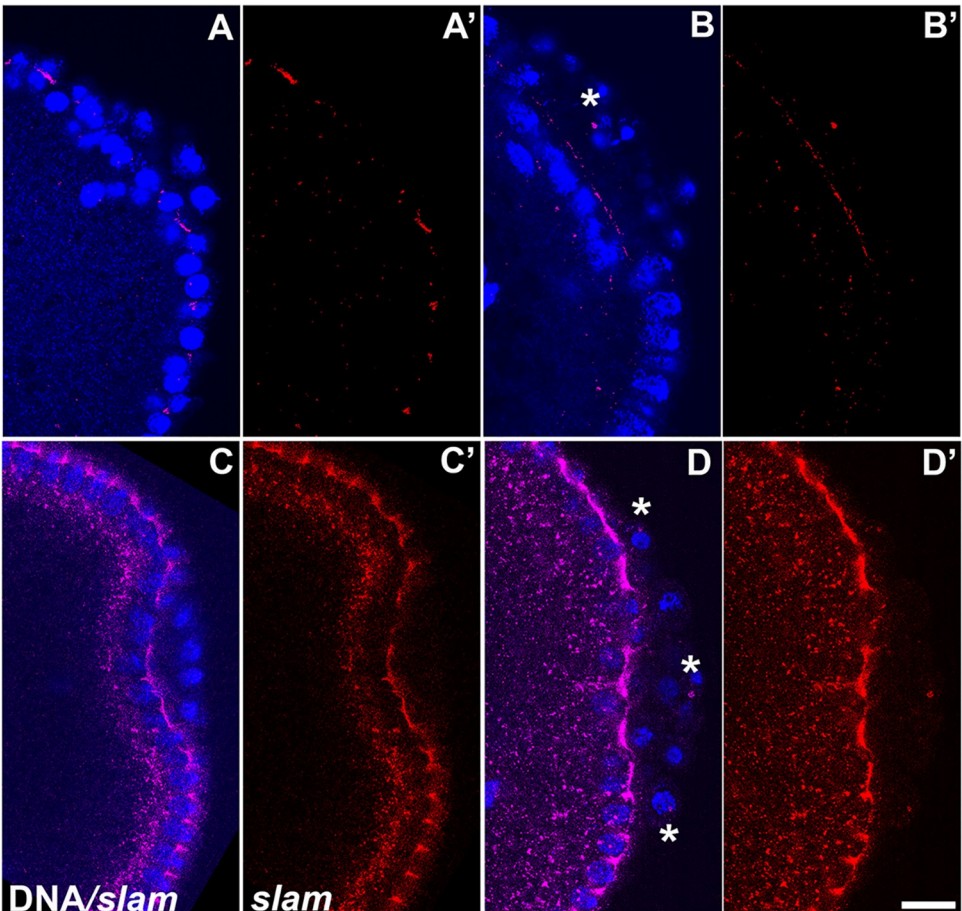

**Fig 5. PGCs from *dpp*-compromised embryos display ectopic expression of the somatic cellularization gene *slow as molasses* (*slam*) in PGCs.** smFISH was performed using probes specific for *slam* (red) on (A/C) *egfpi*, (B) *dpp^{hr92}*, and (D) *dppi^{33618}* embryos. Nuclei were labeled using Hoescht (blue). Asterisks highlight PGC nuclei in *dpp*-compromised embryos expressing *slam* RNA. Scale bar represents 10 μm.

## Compromised BMP signaling correlates with inappropriate distribution of pole plasm components

In analyzing PGCs in embryos compromised for BMP signaling, we noted that Vasa protein was not always fully sequestered in newly formed PGCs (S5 Fig, caret). This led us to wonder if there were defects in the distribution of pole plasm constituents during PGC formation. To explore this possibility, we used smFISH to examine the localization of two pole plasm mRNAs, *pgc* and *gcl*, in *dpp^{hr92}* and *dppi^{33618}* embryos. Fig 6 shows maximum intensity projections of *pgc* mRNA in WT (*egfpi*), *dpp^{hr92}*, and *dppi^{33618}* embryos. In control embryos, *pgc* mRNA is efficiently incorporated into PGCs when they cellularize, and little, if any, escapes to the surrounding soma (Fig 6A). In contrast, in both *dpp^{hr92}* and *dppi^{33618}* pre-cellular blastoderm embryos, incorporation of *pgc* mRNA into PGCs seems less efficient, and *pgc* mRNA is observed in the surrounding soma where it is associated with somatic nuclei (Fig 6B and 6C). The failure to properly sequester *pgc* mRNA in the newly formed PGCs is documented in the plots of individual embryos (Fig 6D). *gcl* mRNA is also not captured efficiently by PGCs in *dpp^{hr92}* and *dppi^{33618}*.

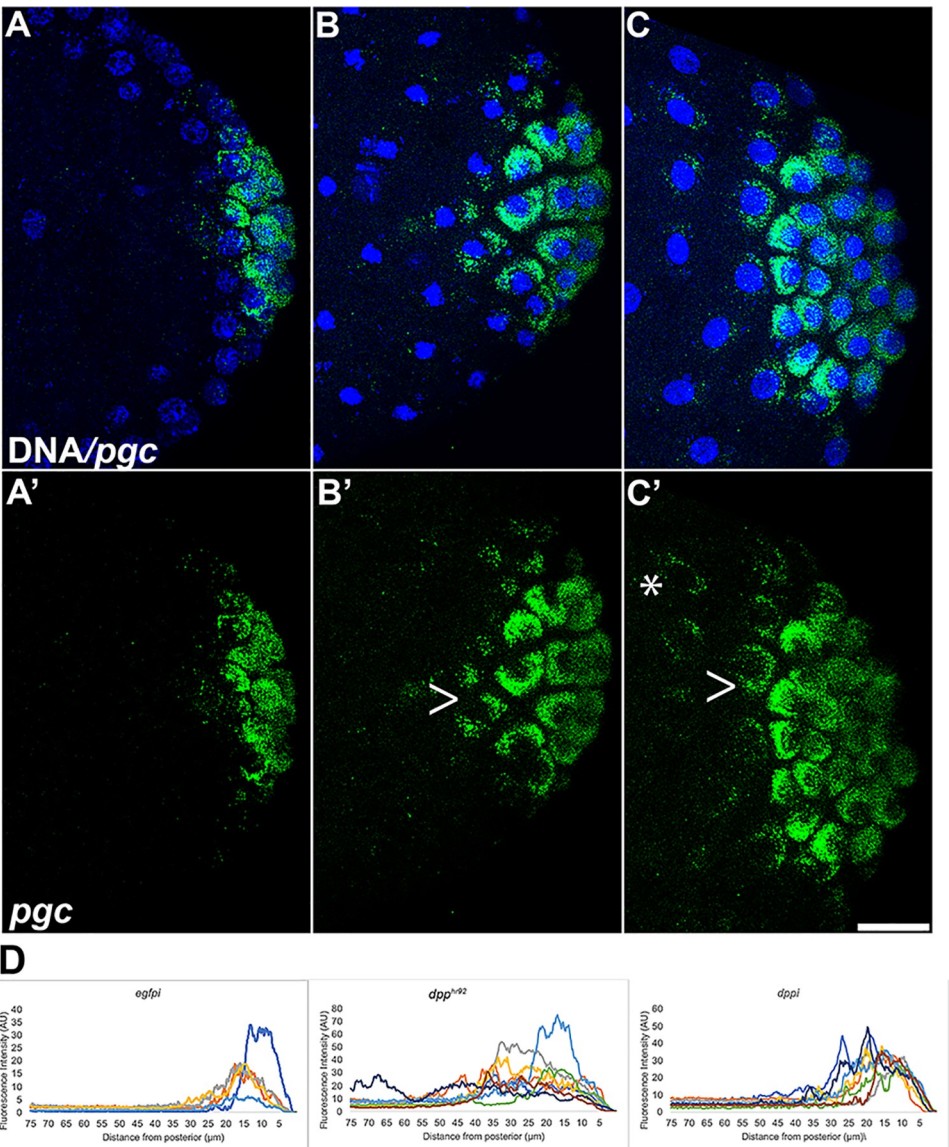

**Fig 6. Loss of *dpp* results in spread of the germ plasm RNA *polar granule component* (*pgc*) away from the posterior pole.** smFISH was performed using probes specific for *pgc* (green) on 0–4 hr paraformaldehyde-fixed (A) *egfpi* (B) *dpp^hr92^* and (C) *dppi^33618^* embryos. Nuclei were labeled using Hoescht (blue). Images shown are representative maximum intensity projections. Scale bar represents 10 μm. (D) Plot profiles showing mislocalization of pole plasm (visualized using *pgc*) away from posterior cap (see Materials and Methods for details of quantification).

The defects in sequestering pole plasm constituents during PGC formation could arise from a defect in the anchoring of the pole plasm RNAs and proteins to the posterior cortex and/or the premature release of these factors. Such defects should be apparent in pre-syncytial embryos prior to nuclear migration into the pole plasm. To assess this possibility, we analyzed the localization of *pgc* and *gcl* RNAs in young embryos before pole cell formation. Unlike in later stages, we found that *pgc* and *gcl* mRNAs are tightly associated with the posterior cortex in all young embryos prior to pole bud formation (n = 5 for each genotype; S6 Fig). Taken together, these data argue that BMP signaling is required for proper sequestration of pole plasm components in PGCS at step during, or more likely after, the release of the pole plasm from the posterior cortical surface.

## BMP signaling is required to suppress the functioning of the terminal pathway

One of the characteristic phenotypes in embryos from *gcl* mutant mothers is a failure to properly sequester pole plasm components during PGC cellularization. As is observed when expression of Dpp is disrupted, *pgc* mRNA and Vasa protein are not properly incorporated into *gcl* mutant PGCs [12]. The failure to properly sequester the pole plasm during PGC formation appears to be due, at least in part, to the inappropriate activation of the terminal signaling pathway in *gcl* mutants. Studies by Pae *et al*. (2017) [65] showed that Gcl targets the terminal pathway receptor Torso for proteolytic degradation, thereby shutting down the pathway at the very posterior pole of the embryo. When the terminal signaling cascade is activated at the posterior pole, either in *gcl* mutants or by gain-of-function mutations in the Torso receptor or its downstream kinases, pole plasm components are not properly incorporated into PGCs when they cellularize [12]. These observations raise the possibility that the BMP pathway might be needed to help suppress the terminal signaling pathway. To test this directly, we probed WT and *dpp^{hr92}* embryos with antibodies against the di-phosphorylated form of the downstream ERK kinase, dpERK, which has been used as a diagnostic marker of pathway activation. WT PGCs have very little, if any, dpERK. In contrast, PGCs in *dpp^{hr92}* embryos accumulate readily detectable levels of dpERK (S7 Fig). We found that 66.0% (n = 47; p = 0.0 by Fisher's exact test) of the PGCs in syncytial blastoderm stage *dpp^{hr92}* embryos have dpERK, while dpERK is found in only 9.8% of the PGCs of equivalently staged WT embryos (n = 51).

## Activation of the terminal pathway at the posterior induces Vasa loss

The presence of dpERK in PGCs when the BMP signaling pathway is compromised indicates that one of its functions in PGC specification is suppressing terminal signaling. To provide further evidence for this connection, we examined Vasa accumulation in PGCs in which the terminal signaling pathway was ectopically activated. For this purpose, we used a light activated SOS protein, *optoSOS*, to turn on the terminal pathway at the posterior pole of the embryo [66, 67]. Fig 7 shows that Vasa protein levels in PGCs are substantially reduced after *optoSOS* activation. Quantification of the average level of Vasa in WT (n = 30) and *optoSOS* (n = 41) late syncytial and cellular blastoderm embryos indicates that there is almost a 2-fold reduction after light activation. This finding, taken together with previous studies (67) which showed

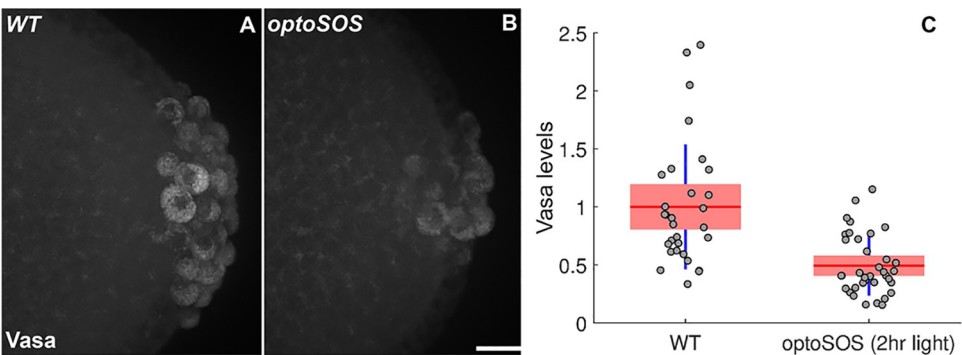

**Fig 7. Optogenetic activation of ERK signaling in the embryonic posterior results in loss of Vasa from PGCs.** Paraformaldehyde-fixed and blue-light exposed (A) *WT* (n = 30) and (B) *OptoSOS* (n = 41) NC12-14 embryos were stained for the pole cell marker Vasa (white). Images shown are representative maximum intensity projections. Scale bar represents 10 μm. (C) Normalized total Vasa intensities were measured. Red line shows the mean, red box indicates the 95% confidence interval, and blue lines indicate standard deviation (see Materials and Methods for details of quantification).

that *pgc* and *gcl* mRNAs are not properly incorporated into PGCs unless the terminal signaling pathway is shutdown during their cellularization, would provide additional support for the idea that this pathway is an important target for the BMP signaling pathway in PGC specification. On the other hand, it seems likely that the terminal pathway is not the only BMP target: we found that accumulation of dpERK either via optogenetic activation or expression of a constitutively active form of upstream kinase does not lead to inappropriate and/or precocious activation of somatic target genes such as *tll* [68].

## Loss of *dpp* results in precocious invasive migration of PGCs

Hyperactivation of the terminal pathway in late syncytial or early cellular blastoderm embryos is known to induce an unusual change in the behavior of PGCs. Instead of remaining adhered to one another in a monolayer on the surface of the embryo, a subset of the PGCs lose adhesion and begin to invade the underlying soma. Since loss of BMP signaling induces ERK phosphorylation in PGCs, one might expect to observe evidence of invasive migration. As indicated in S7 Table, invasive migration is observed only infrequently in *dpp^{hr92}* embryos. Invading PGCs were observed in only 13.3% of *dpp^{hr92}* embryos (S7 Table; p = 0.484127 by Fisher's exact test). On the other hand, in the case of the strong hypomorph allele, *dpp^4*, invasive migration was observed in nearly one half of the *dpp^4* embryos (Fig 8 and S7 Table, p = 0.014907 by Fisher's exact test).

## Dominant genetic interactions between *osk* and *dpp*

Our findings demonstrate that the early steps in PGC specification are disrupted when the functioning of the BMP signaling pathway is compromised. Factors important for the specification process are not properly segregated into newly formed PGCs, and critical steps in establishing PGC identity such as shutting down the terminal pathway and imposing transcriptional quiescence are disrupted. These steps are also known to be dependent upon the *osk* gene, which encodes the maternal determinant that orchestrates the assembly of the pole plasm during late stages of oogenesis [44, 45]. In this case, one might anticipate that there would be synergistic genetic interactions between *osk* and the BMP pathway even though their critical functions are required in different contexts—*osk* in the mother and BMP signaling in the zygote.

To test this prediction, we reduced *osk* activity in the mother and BMP signaling in the zygote by mating *osk^{A87}/+* heterozygous mothers to *dpp^{hr92}* homozygous fathers. All the embryos derived from this cross i.e. "*dpp^{hr92}/+; osk/+*" have a WT *dpp* gene trans to the

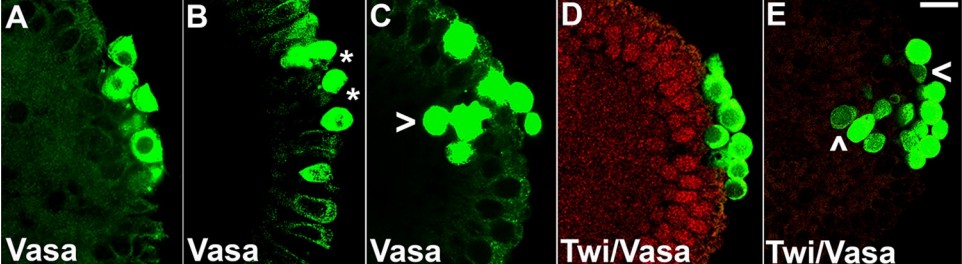

**Fig 8. Loss of *dpp* function results in invasive migration and lack of adherence of PGCs.** 0–4 hr paraformaldehyde-fixed (A) *osk/+* (B/C) *dpp^{hr92}/+;osk/+* (D) WT and (E) *dpp^4* embryos were stained for Vasa (A-E, green) and Twist (D-E, red). Asterisks show spread of PGCs/lack of adherence in *dpp^{hr92}/+;osk/+* embryos while carets highlight invasive migration in *dpp^{hr92}/+;osk/+* and *dpp^4* embryos. Images are representative maximum intensity projections. Scale bar represents 10 μm.

hypomorphic (and homozygous viable) *dpp^hr92* allele. As a control, we used embryos obtained from *osk^A87*/+ mothers (*osk*/+) mated to WT males. We expected that if *dpp* signaling functions in collaboration with the *osk*-dependent specification pathway, simultaneous reduction in the activity of both may impact the formation and/or specification process.

We found no evidence that PGC formation was altered in *dpp^hr92*/+; *osk*/+ embryos. The average number of PGCs observed in *dpp^hr92*/+; *osk*/+ was not significantly different from that in *osk*/+. Cycle 14 *dpp^hr92*/+; *osk*/+ embryos had 11.7 PGCs (n = 49) while the *osk*/+ embryos had 11.1 PGCs (n = 37) (S8 Table; p = 0.49 by t-test).

While PGC formation appeared to be normal, there were defects in specification. While the accumulation and distribution of Vasa is perturbed in both *osk*/+ and *dpp^hr92*/+; *osk*/+, the frequency of defects are greater in *dpp^hr92*/+; *osk*/+ (Fig 8A, 8B, and 8C). In *osk*/+ embryos, 42.3% (n = 26) had reduced levels of Vasa, while in *dpp^hr92*/+;*osk*/+ embryos, reductions in Vasa were evident in 69.2% (n = 26) of the PGCs (S9 Table; p = 0.092935 by Fisher's exact test). The loss of Vasa is likely due, at least in part, to the failure to properly segregate germ plasm components into the PGCs during the process of cellularization. In this case, we would expect to observe the aberrant localization of other germ plasm components into the surrounding soma. Fig 9 shows the localization of *pgc* mRNAs in *osk*/+ and *dpp^hr92*/+;*osk*/+ embryos. The efficiency of incorporation of *pgc* mRNAs into *osk*/+ PGCs in this experiment (Fig 9A, 9C, and 9E) appears similar to WT (Fig 6A). In contrast, *pgc* mRNAs are not properly captured by the PGCs in *dpp^hr92*/+;*osk*/+ embryos (Fig 9B, 9D, and 9F) and instead spread into the soma. To extend these observations, we simultaneously hybridized *osk*/+ and *dpp^hr92*/+;*osk*/+ embryos

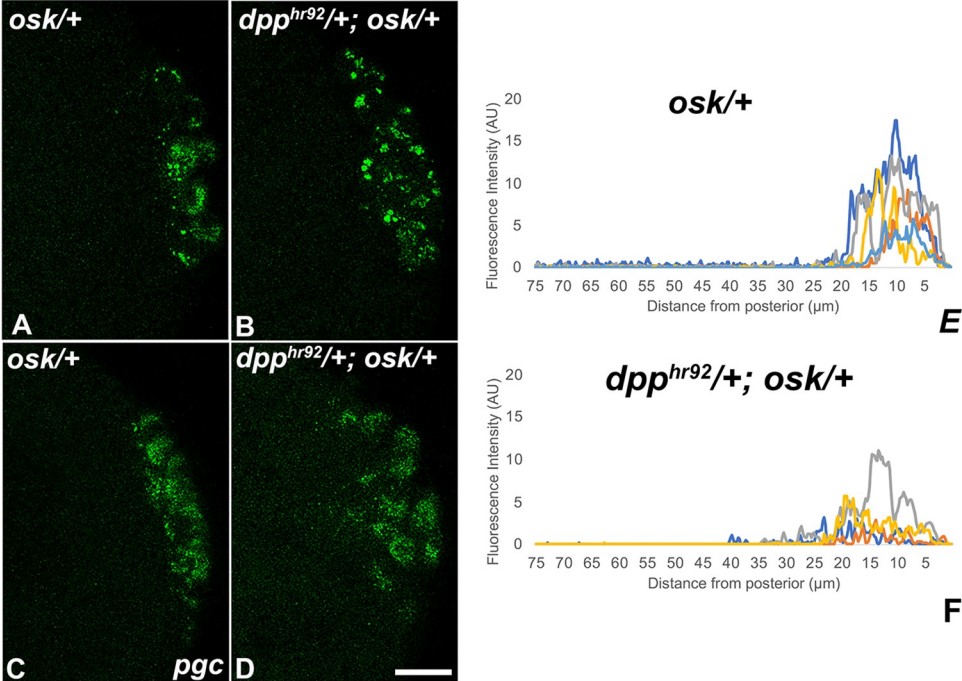

**Fig 9. Simultaneously compromising maternal and zygotic levels of *oskar* and *dpp* respectively results in aberrant localization of germ plasm.** smFISH was performed using probes specific for *pgc* (green) on 0–4 hr paraformaldehyde-fixed (A/C) *osk*/+ and (B/D) *dpp^hr92*/+;*osk*/+ embryos to assess spread of pole plasm from the posterior. Images are representative maximum intensity projections Scale bar represents 10 μm. (E-F) Plot profiles show mislocalization of pole plasm (visualized using *pgc*) away from posterior cap (see Materials and Methods for details of quantification). Each plot shows a representative experiment, with each line depicting pole plasm distribution of an individual embryo.

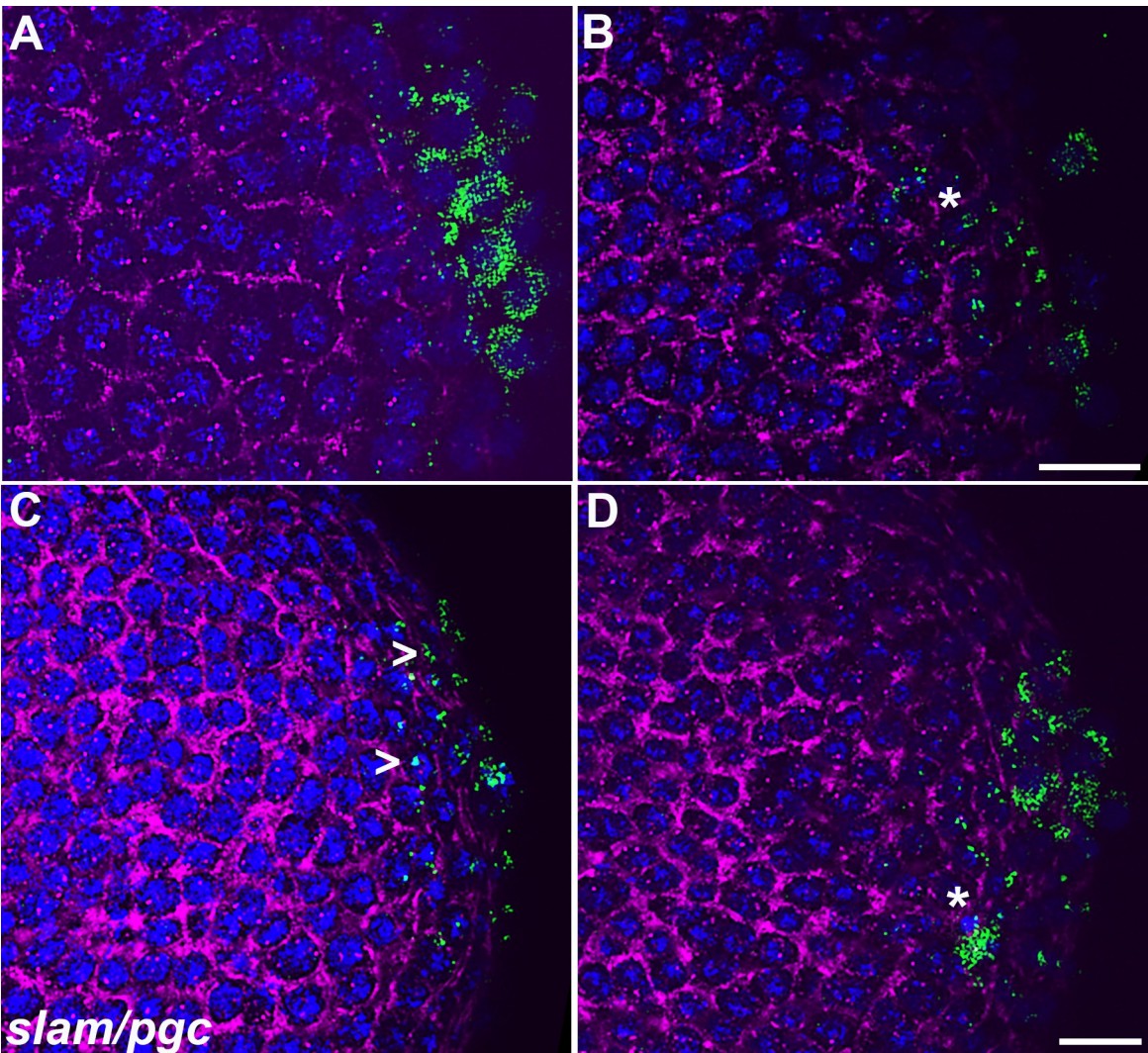

**Fig 10. Aberrantly transmitted *pgc* RNA colocalizes with somatic marker, *slam* upon simultaneous compromise of *oskar* and *dpp*.**
smFISH was performed using probes specific for *pgc* (green) and *slam* (magenta) on 0–4 hr paraformaldehyde-fixed (A/C) *osk*/+ and (B/D) *dpp*^hr92^/+;*osk*/+ embryos to assess spread of pole plasm from the posterior. Caret indicates that some somatic nuclei are exposed to *pgc* in *osk*/+ embryos. Asterisk highlights the more severe mislocalization of *pgc* and overlap of *pgc* and *slam* signals in *dpp*^hr92^/+;*osk*/+ embryos. Images are representative maximum intensity projections Scale bar represents 10 μm.

with *slam* and *pgc* mRNA probes. As shown in Fig 10B and 10D, *pgc* mRNA is found associated with somatic nuclei that are expressing *slam* mRNAs (see asterisks). In this experiment, we also occasionally observed *pgc* mRNA associated with somatic nuclei expressing *slam* mRNA in the *osk*/+ control embryos (Fig 10A and 10C, caret)). However, this occurs much less frequently, indicating that mislocalization of *pgc* mRNA is greatly exacerbated by a modest reduction in *dpp* activity.

After PGC formation and their initial specification, they remain adhered to one another and cluster together in a relatively uniform monolayer overlying the posterior pole of syncytial and cellular blastoderm embryos. While this is also true for most of the *osk*/+ control embryos, we observed striking defects in PGC grouping at the posterior pole in *dpp*^hr92^/+;*osk*/+ embryos, presumably from loss of adhesion (Figs 8B and S8). 56.7% of these embryos (n = 78; p = 0.000151 by Fisher's exact test) have one or more PGCs that are detached from the

posterior cluster of PGCs, as compared to only slightly more than a quarter of the *osk*/+ embryos (26.7% (n = 75)) (S10 Table). In addition to lack of clustering, $dpp^{hr92}$/+;*osk*/+ PGCs do not always remain on the surface of the embryo. Instead, they begin invading the underlying soma. An example of invasive migration by the PGCs in $dpp^{hr92}$/+;*osk*/+ embryos is shown in Fig 8C. The invasive phenotype was observed in 38.5% (n = 78; p = 0 by Fisher's exact test) of $dpp^{hr92}$/+;*osk*/+ embryos. By comparison, invasive PGCs were detected in only 5.3% of the *osk*/+ control embryos (n = 75) (S10 Table).

## Discussion

*Drosophila melanogaster* is one of the classic and most thoroughly studied examples of organisms that use a "preformation" mechanism for PGC specification [1]. One of the first indications that PGCs in flies are specified by preformation came from experiments nearly 100 years ago in which the germline determinants were inactivated by UV irradiating the posterior of the embryo [69–71]. The existence of localized and UV-sensitive determinants was subsequently supported by transplantation experiments. Okada *et al.* showed that the sterility of UV irradiated embryos could be rescued by injection of cytoplasm from unirradiated embryos into the posterior region of the UV treated embryos [72]. Importantly, they found that rescuing activity is localized within the donor; it is present at the posterior, but not the anterior pole. In complementary experiments, cytoplasm from the posterior pole was found to induce PGCs when injected into the anterior [43]. Moreover, these ectopic PGCs were able to generate a functional germline if reinjected into the posterior pole of similarly staged recipients. Subsequent experiments showed that both PGC formation and abdominal patterning requires genes that are active in the mother during oogenesis and that the key maternal factor for the formation of the pole plasm is encoded by *osk* [41, 42, 44, 73]. Ectopic localization of *osk* mRNAs engineered by linking it to *bcd* 3' UTR is sufficient for the assembly of pole plasm at the anterior of the oocyte [44]. In pre-cellular blastoderm embryos, the ectopic pole plasm induces the formation of PGCs at the anterior end. Moreover, as was first observed by Illmensee and Mahowald [43], these PGCs are able to populate the adult germline if transplanted into the posterior of recipient embryos.

While these and other studies [45] clearly demonstrate that maternal determinants localized at the posterior pole of the fly embryo orchestrate PGC specification, they do not establish that the mechanism is exclusively preformation. There is a potentially significant caveat with the experiments showing that PGCs induced at the anterior (either by injection of pole plasm or by the *osk-bcd* 3' UTR) are capable of generating a functional adult germline: the ectopic PGCs were transplanted into the posterior of recipient embryos and thus are subject to much the same milieu as PGCs formed by the normal mechanisms. This leaves open the possibility that epigenesis might play a role in PGC specification. To address this issue, we have asked whether the BMP signaling pathway is involved in the specification of fly PGCs when they are formed during the pre-cellular blastoderm stages of embryogenesis. We selected this pathway for four reasons. First, this pathway is known to function in PGC specification in animals that rely on epigenesis [1]. Second, though *dpp* expression in the zygote is restricted to the dorsal side of the embryo by the Dorsal morphogen, both *in situ* hybridization and antibody staining experiments indicate that *dpp* expression encompasses the entire posterior pole [56]. Third, experiments by Dorfman and Shilo showed that the transcriptional activator, pMad, induced in receiving cells by the BMP signaling pathway is present in the nuclei of pre-cellular blastoderm PGCs [52]. Moreover, pMad accumulation depends not only on the Dpp ligand, but also maternal *and* zygotic sources of the Tkv receptor. Lastly, in previous studies we found that

BMP signaling is required to maintain PGC identity and their differentiation during mid-embryogenesis in the period leading up to the coalescence of the embryonic gonad [50].

Our results show that maternal determinants are insufficient on their own for proper specification of PGCs and that this process is not exclusively cell autonomous as has long been thought. Instead, a hybrid of preformation and epigenesis is deployed to generate a full complement of functional PGCs. We find that when the BMP pathway is disrupted in pre-cellular blastoderm embryos, newly formed PGCs exhibit a variety of defects indicative of a failure in PGC determination. Moreover, the same phenotypes are observed in PGCs in embryos whose mothers are mutant in one or more of the three maternal factors, *gcl*, *pgc* and *nos*, that are known to be required for proper PGC specification. It is also worth noting that the functioning of the BMP pathway was not completely disrupted in our experiments. Thus, the possibility remains open that more dramatic or even some additional phenotypic effects might be observed under conditions where BMP signaling was entirely absent in pre-cellular blastoderm embryos. (For instance, *dpp* activity may affect PGC cellularization. It will also be of interest to assess if pole plasm anchoring is completely normal in pre-blastoderm embryos if BMP pathway is compromised).

Included in the defects that were observed are the partial loss of Vasa protein, a failure to downregulate Ser2 CTD phosphorylation, changes in the profile of histone modifications and the phosphorylation of the terminal (and EGFR) signaling pathway protein ERK. In WT PGCs, phosphorylation of Ser2 in the CTD domain of the large PolII subunit by pTfb is blocked by the Pgc protein [13–15]. This block is overridden in embryos homozygous for the partial loss of function *dpp^{hr92}* allele and in embryos produced by *tkv⁻* germline clone mothers. Moreover, two of the known targets for Pgc repression, *tll* and *slam*, are expressed in *dpp^{hr92}* PGCs. However, the misfunctioning of Pgc is not the only defect in establishing transcriptional quiescence. We also observed activation of the *Sxl* establishment promoter, *Sxl-Pe*. Previous studies have shown *Sxl-Pe* transcription is inappropriately turned on in PGCs in the progeny of *gcl* and *nos* mothers, but not in the progeny of *pgc* mothers [12, 13, 17]. Thus, compromising BMP signaling would seem to broadly impact transcriptional quiescence, leading to the misexpression of genes normally repressed in PGCs by the activity of several different factors. In the case of *tll*, for example, Pgc is not the only pole plasm component expected to play a role in its repression. *tll* transcription is activated in somatic nuclei at the anterior and posterior ends of the embryo by the terminal pathway. However, the terminal pathway is normally shut down in WT PGCs by Gcl protein, which mediates the degradation of the Torso receptor [65]. *gcl* function must also be disrupted either directly or indirectly when the BMP signaling pathway is compromised as we find that dpERK accumulates in PGC nuclei just as in *gcl* mutants.

These are not the only connections between the BMP pathway, Gcl, and the terminal signaling pathway. In *gcl* embryos, components of the pole plasm (including Vasa and *pgc*) are not properly captured by PGCs when they cellularize. Instead, they spread into the posterior region of the embryo and are found associated with somatic nuclei. A similar disruption in the proper distribution of pole plasm during PGC cellularization was found when the terminal pathway was upregulated by blocking degradation of Torso, or by a constitutive allele of the downstream MEK kinase [12]. Moreover, as shown here, we find that Vasa is lost from PGCs when the terminal pathway is optogenetically activated at the posterior of the embryo. In fact, the Vasa loss phenotype induced by optogenetic activation of SOS is quite similar to that seen when we disrupted the BMP pathway by RNAi knockdown of *dpp* and by *dpp* and *tkv* mutants. Taken together, these findings argue that one of the functions of the BMP pathway in PGC specification is to block the terminal signaling pathway. In this context, it is interesting to note that inhibition of EGFR-dependent signaling may be important for PGC specification in mammals. When ESC cells are cultured under conditions that promoted the formation of

mesoderm lineages, a PGC-like identity could be induced by adding an inhibitor of the upstream kinase MEK to the culture media [74].

In flies there is a complicated relationship between the BMP and EGFR signaling pathways. In the case of the Torso-dependent terminal pathway, dpERK phosphorylation of Capicua in the anterior and posterior soma counteracts the repression of *dpp* by Dorsal [75, 76]. In other developmental contexts, the relationship between BMP signaling and the cognate EGFR pathway is complex. In the wing disc, for example, BMP and EGFR signaling is reported to establish a positive feedback loop, reinforcing each other by promoting the synthesis of their respective ligands [77]. In other contexts, however, the interactions between the two pathways is different. Genome wide studies of dorsoventral patterning during embryogenesis indicate that BMP signaling both negatively and positively regulates the expression of components of the EGFR signaling pathway during embryogenesis [78]. EGFR, in turn, was proposed to temper rather than augment the BMP pathway by a dpERK-dependent phosphorylation of dSmad that results in its degradation. In studies on the patterning of the eye field and head epidermis in *Drosophila*, Chang *et al.* also proposed an antagonistic relationship between BMP and EGFR signaling [79]. High levels of Dpp were found to block EGFR signaling by inhibiting dpERK accumulation, while EGFR gain of function mutants suppress Dpp signaling. Though the mechanism for inhibiting dpERK accumulation was not uncovered, the same mechanism could be deployed to block the activation of the terminal pathway during PGC specification in pre-cellular blastoderm embryos. Alternatively, the mechanism of inhibition may be specific to the process of PGC specification. For example, studies by Pae *et al.* (2017) would predict that Torso degradation by Gcl should in itself be sufficient to eliminate both canonical and non-canonical activities of the terminal signaling pathway in PGCs [65]. Thus, it is possible that the BMP pathway might inhibit terminal signaling by potentiating Gcl activity either directly or indirectly.

That the BMP signaling pathway is required for the proper functioning of maternally deposited pole plasm components during PGC specification is also suggested by the dominant genetic interactions between *osk* and *dpp*. In these experiments, females heterozygous for an *osk* mutation were mated to males carrying the weak viable *dpp* allele, *dpp*^hr92^. Reducing the dose of *osk* in the mother by itself appears to result in a minor perturbation of pole plasm sequestration in her progeny; however, this defect is substantially enhanced when the progeny are also heterozygous for *dpp*^hr92^. In addition to failing to completely capture the pole plasm, *dpp*^hr92^/+;*osk*/+ PGCs exhibit other abnormalities, including a novel loss of cell:cell adhesion and invasive migration. This synergistic interaction would argue that the BMP pathway collaborates with *osk* in the process of PGC specification, and in doing so serves to integrate preformation with epigenesis.

The fact that one of the classical models of preformation deploys a signaling pathway that is known to play a critical role in PGC specification in species that rely on epigenesis would seem to bolster the argument that epigenesis is the ancestral mode for generating this special cell identity. This view would be supported by the evolutionary history of *osk* and *nos*, genes that function cell autonomously in PGC specification in flies. The former is restricted to a subset of insects that utilize preformation in PGC specification and is thought to arise from the fusion of bacterial and eukaryotic sequences [80]. *nos*, by contrast, is conserved from worms to human and spans species that are classically identified as using either preformation or epigenesis for PGC specification.

## Materials and methods

### Fly stocks and genetics

The following fly stocks were used for the analysis reported in this manuscript. *white*^1^ (*w*^1^) was used as the WT stock. *dpp*^4^*twi* recombinant stock was obtained from Eric Wieschaus. *egfp*

*RNAi* (BDSC #41552), *dpp RNAi* lines (BDSC #33618 or 25782), and *UAS-dpp* (BDSC #1486) were driven by either *twi-GAL4* driver stock (BDSC #2517) or *maternal-tubulin-GAL4* (*67.15*) driver stock, which carries 4 copies of *maternal-tubulin-GAL4* (obtained from from Eric Wieschaus). BMP signaling was also compromised using *dpp$^{hr92}$* (BDSC #2069), tkv$^{427}$ (obtained from Kristi Wharton), or *FRTtkv$^8$* stock (obtained from Michael O'Connor). *osk$^{A87}$* stock was a gift of Liz Gavis.

For the embryo staining, we used *white$^1$* embryos as a control in most instances. Wherever the experiment involved an RNAi knockdown strategy, we generated *egfpi* embryos by crossing *maternal-tubulin-GAL4* virgins with *UAS-egfpi* males and used these as control.

## Immunostaining

Embryos were formaldehyde-fixed, and a standard immunohistochemical protocol was used for fluorescent- or DAB-visualized immunostaining as described previously [17]. The primary antibodies used were rabbit anti-Vasa (1:2000; gift of Paul Lasko), rat anti-Vasa (1:1000; gift of Paul Lasko), rabbit anti-pH3 (1:1000; Upstate Biotechnology), rabbit anti-CycB (1:500; gift of Jordan Raff), mouse anti-H5 to detect pSer2 (1:250; Research Diagnostics, Inc.), H3meK4 (1:500; gift of C. David Allis), rat anti-Twist (1:500; gift of Eric Wieschaus), rabbit anti-dpERK (1:100; Cell Signaling Technology), sheep anti-GFP (1:1,000; Bio-Rad), sheep anti-digoxigenin (DIG) (1:125; Roche), and mouse anti-biotin (1:125; Jackson Immunoresearch). Fluorescent immunostaining employed Alexa-Fluor secondary antibodies used at 1:500 (ThermoFisher), and DNA was labeled using either DAPI (10 ng/mL, ThermoFisher Scientific) or Hoescht (3μg/ml, Invitrogen). For DAB staining, horse radish peroxidase (HRP) secondary antibodies (Jackson Immunoresearch) were used 1:1000. Stained embryos were mounted using Aqua Poly/mount (Polysciences) on slides. At least three independent biological replicates were used for each experiment.

## Fluorescence *in situ* hybridization (FISH) and single molecule FISH (smFISH)

FISH was performed as previously described using probes specific to *tll* [68]. To quantify *tll* levels, we used an internal control within each embryo to normalize the intensity. Using ImageJ, we measured and averaged intensities of three randomly selected PGCs, and we also collected averaged intensities from three measurements per embryo within the somatic cells positive for *tailless*. The normalization was done according to the following: *Normalized intensity = Averaged intensity from PGCs / averaged intensity from somatic cells*. This normalization was done for each condition separately. These normalized intensities for individual embryos were plotted and presented in Fig 4.

smFISH was performed as previously described using formaldehyde-fixed embryos [12]. All probe sets were designed using the Stellaris probe designer (20-nucleotide oligonucleotides with 2-nucleotide spacing). *pgc* and *gcl*, smFISH probes (coupled to either atto565 or atto647 dye, Sigma) were a gift from Liz Gavis. *Sxl-Pe* intronic probes (coupled to atto565 dye) were a gift from Thomas Gregor., and *slam* probes (coupled to Quasar 670) were produced by Biosearch Technologies. All samples were mounted using Aqua Poly/mount (Polysciences) on slides. At least three independent biological replicates were used for each experiment.

## Optogenetic activation of ERK

OptoSOS and Hist-GFP WT control embryos were collected in the dark for 2 hours and then stimulated with blue light. Blue light stimulation was done at ~1mW/cm2 at 450 nm for 2 hours using a custom-built panel of 30 LEDS placed ~5cm from the embryos and enclosed in

foil. After stimulation, Hist-GFP WT control and OptoSOS embryos were immediately pooled, decorated, and placed in fixative. During fixation embryos remained in the blue light for 10 minutes to ensure continued activation. After fixation, embryos were stained for Vasa. Z-stacks were taken of NC12-14 embryos (WT n = 30, OptoSOS n = 41). Images were processed in MATLAB, k-means clustering with 4 bins was applied to the max-projected Vasa images to segment the vasa positive cells (using the highest bin). The background staining of the embryo was calculated using the mean intensity of the middle 2 bins. This background was subtracted from the average intensity of the segmented vasa positive cells. These averages were calculated for each embryo in the optogenetic and hist-GFP groups before normalizing both averages by the mean of the hist-GFP group.

## Microscopy and image analysis

NIKON-Microphot-SA microscope was used to capture images of DAB-stained embryos (40X). Imaging for all other smFISH and fluorescent immunostaining experiments was performed on a Nikon A1 inverted laser-scanning confocal microscope.

Images were assembled using ImageJ (NIH) and Adobe Photoshop and Illustrator software to crop regions of interest, adjust brightness and contrast, generate maximum-intensity projections, and separate or merge channels. To assess the mislocalization of the RNAs or protein in different genetic backgrounds compared to the control, we generated plot profiles using ImageJ. The posterior-most 75 μm of each embryo was plotted for comparison, and embryos from a single biological replicate are plotted in figures given that variation between fluorescence between replicates obscured the pole plasm distribution trends if embryos from all replicates were plotted together.

## Statistical analysis

Using NC13/14 embryos, PGCs of each genotype were counted from the 1st Vasa-positive cell to the last through an entire z-volume captured at 2 micron intervals. These PGCs counts were analyzed using a Student's t-test. The same analysis was applied when counting either pH3- or CycB-positive PGCs in individual embryos for *dpp* gain of function experiments.

To compare numbers of PGCs with high or low levels of listed markers (Vasa, pSer2, H3meK4, dpERK), PGCs either positive or negative for each marker were counted from all identifiable PGCs of each embryo. The differences in marker levels were consistent, and PGCs could be easily classified into each category. Individual PGCs were counted by going through each slice of a Z stack and categorizing each cell (viewed throughout all relevant slices), and pairwise comparisons of these populations for each genotype were performed using Fisher's Exact test. For FISH/smFISH experiments, total number of embryos expressing *tll*, *slam*, or *Sxl-Pe* in PGCs were counted, and Fisher's Exact Test was used to test significance in the compared proportions of embryos positive for transcription in PGCs. Likewise, proportions of embryos displaying aberrant PGC behavior (lack of adhesion/invasion) were compared to control embryos using Fisher's Exact Test. Data were plotted and statistical analyses were performed using Microsoft Excel or R Project software.

## Supporting information

**S1 Fig. *dpp* gain of function embryos have increased number of PGCs.** 0–4 hr paraformaldehyde-fixed (A/C) *twi-Gal4* (B/D) or *twi-Gal4/UAS-dpp* stage 4 (A-B) or stage 5 (C-D) embryos were stained for pole cell marker Vasa (red) to assess pole cell number and proliferation. Asterisks indicate additional divisions. Scale bar represents 10 μm.
(JPG)

**S2 Fig. PGCs from *dpp* gain of function embryos are elevated in number but retain transcriptional quiescence.** 0–4 hr paraformaldehyde-fixed (A) *twi-Gal4* (B) or *twi-Gal4/UAS-dpp* embryos were stained for pole cell marker Vasa (red) and phosphoSer2 (transcriptional activation, green). Asterisk indicates excess division shown by greater number of Vasa-positive cells. Carets shows lack of pSer2 in PGCs, suggesting transcriptional quiescence. Scale bar represents 10 μm.
(JPG)

**S3 Fig. Young *tkv* germline clone embryos display marker of transcriptional activation in PGCs.** 0–4 hr paraformaldehyde-fixed (A) *WT* and (B/C) *tkv^m-^* embryos were stained for pole cell marker Vasa (red) and phosphoSer2 (transcriptional activation, green). Caret highlights absence of pSer2 signal, suggesting transcriptional quiescence, while asterisk shows that loss of Vasa correlates with gain of transcriptional activation. Scale bar represents 10 μm.
(JPG)

**S4 Fig. Disrupted BMP signaling results in patchy accumulation of H3meK4 in PGCs.** 0–4 hr paraformaldehyde-fixed (A) *WT* and (B) *tkv^421^* embryos were stained for the pole cell marker Vasa (blue) and H3meK4 (red). Caret shows absence of H3meK4 while asterisk highlights aberrant presence of H3meK4 signal in PGCs, a marker of transcriptionally active chromatin. Scale bar represents 10 μm.
(JPG)

**S5 Fig. Compromising *dpp* levels results in ectopic localization of Vasa from posterior pole of embryos.** 0–4 hr paraformaldehyde-fixed (A) *egfpi* and (B) *dpp^hr92^* embryos were stained for pole cell marker Vasa (green). Nuclei were labeled using Hoescht (blue). Asterisk shows Vasa localization away from posterior pole and invasive migration of PGCs. Scale bar represents 10 μm. (C) Plot profiles showing mislocalization of pole plasm (visualized using Vasa) away from posterior cap (see Materials and Methods for details of quantification).
(JPG)

**S6 Fig. Loss of *dpp* does not affect germ plasm RNA anchoring at posterior pole prior to PGC budding.** smFISH was performed using probes specific for *pgc* (green) and *gcl* (magenta) on 0–4 hr paraformaldehyde-fixed (A) *egfpi* (B) *dpp^hr92^* and (C) *dppi^33618^* embryos.
(JPG)

**S7 Fig. *dpp*-compromised embryos display dpERK in PGCs.** 0–4 hr paraformaldehyde-fixed (A) *WT* and (B) *dpp^hr92^* embryos were stained for dpERK (red). Nuclei were labeled using Hoescht (blue). Caret indicates lack of dpERK in WT PGCs while the asterisk highlights ectopic dpERK in *dpp^hr92^* PGCs. Scale bar represents 10 μm.
(JPG)

**S8 Fig. Simultaneous compromise of *oskar* and *dpp* reduces adherence between PGCs.** 0–4 hr paraformaldehyde-fixed (A) *osk/+* and (B) *dpp^hr92^/+;osk/+* embryos were stained for the pole cell marker Vasa. Scale bar represents 10 μm.
(JPG)

**S1 Table. PGC counts increase in *dpp* gain of function embryos.**
(XLSX)

**S2 Table. Mitotic markers increase in PGCs of *dpp* gain of function embryos.**
(XLSX)

**S3 Table. The PGC marker Vasa and hallmark of transcriptional activation pSer2 are unaltered in *dpp* gain of function PGCs.**
(XLSX)

**S4 Table. Levels of Vasa decrease in PGCs of embryos compromised for BMP signaling components.**
(XLSX)

**S5 Table. Markers of active transcription increase in embryos compromised for BMP signaling.**
(XLSX)

**S6 Table. Incidence of ectopic PGC transcription of *slam* and *Sxl* in *dpp*-compromised embryos.**
(XLSX)

**S7 Table. Invasive migration of PGCs in *dpp*-compromised embryos.**
(XLSX)

**S8 Table. PGC counts in embryos simultaneously compromised for *osk* and *dpp*.**
(XLSX)

**S9 Table. Vasa levels decrease upon simultaneous compromise of *osk* and *dpp* in blastoderm embryos.**
(XLSX)

**S10 Table. PGC specification and behavior in embryos simultaneously compromised for *osk* and *dpp*.**
(XLSX)

## Acknowledgments

Authors gratefully acknowledge Eric Wieschaus and Trudi Schupbach for many discussions, useful suggestions, and reagents over the course of this work. Liz Gavis, Stas Shvartsman and Jared Tottcher are thanked for continued support. Chris Ng and Gordon Grey provided technical assistance and fly food respectively. We thank Dr. Gary Laevsky and the Confocal Imaging Facility, a Nikon Center of Excellence, in the Department of Molecular Biology at Princeton University for instrument use. G.D. thanks his friends at the Division of Biology from IISER Pune for their generous hospitality.

## Author Contributions

**Conceptualization:** Megan M. Colonnetta, Paul Schedl, Girish Deshpande.

**Formal analysis:** Megan M. Colonnetta, Yogesh Goyal, Heath E. Johnson, Sapna Syal.

**Funding acquisition:** Megan M. Colonnetta, Yogesh Goyal, Heath E. Johnson, Paul Schedl, Girish Deshpande.

**Investigation:** Megan M. Colonnetta, Yogesh Goyal, Heath E. Johnson, Sapna Syal, Girish Deshpande.

**Supervision:** Paul Schedl, Girish Deshpande.

**Writing – original draft:** Megan M. Colonnetta, Paul Schedl, Girish Deshpande.

**Writing – review & editing:** Megan M. Colonnetta, Yogesh Goyal, Heath E. Johnson, Sapna Syal, Paul Schedl, Girish Deshpande.

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
