## [Decision Letter · Decision Letter 0]

16 Sep 2021

Dear Dr Deshpande,

Thank you very much for submitting your Research Article entitled 'Preformation and epigenesis converge to specify primordial germ cell fate in the early Drosophila embryo' to PLOS Genetics.

The manuscript was fully evaluated at the editorial level and by independent peer reviewers. The reviewers appreciated the attention to an important topic but identified some concerns that we ask you address in a revised manuscript

We therefore ask you to modify the manuscript according to the review recommendations. Your revisions should address the specific points made by each reviewer. In particular, please see reviewer #1 point #5. Please acknowledge the caveat of manipulating Dpp signaling in embryos possibly lacks precision and how this impacts interpretation. Although not necessary, you may also wish to address this point with additional experiments.

[LINK]

Yours sincerely,

Giovanni Bosco, Ph.D.

Associate Editor

PLOS Genetics

Gregory P. Copenhaver

Editor-in-Chief

PLOS Genetics

Reviewer's Responses to Questions

**Comments to the Authors:**

Reviewer #1: In this paper, Colonetta et al suggest that Dpp signalling contributes to germ cell formation in Drosophila and frame this suggestion in the context of the classic epigenesis/preformation debate. The authors show that the number of germ cells is somewhat modulated by changes in Dpp signalling (LOF and GOF). They show that this could be mediated in part by an effect on PGC proliferation but then argue, based on changes in vasa expression, that Dpp regulates an early step in PGC specification. They then describe a number of observations suggesting that the PGC state is ‘weakened’ upon loss of Dpp activity.

The paper starts with an intriguing idea, that germ cells specification relies on a combination of epigenesis and cytoplasmic determinants but unfortunately the evidence is somewhat incomplete, dampening initial enthusiasm. In light of the unexpected nature of their finding, there is a particular burden on proof placed on the authors.

Here are some of the issues (both logical and experimental).

1) The author suggest that Dpp could contribute to the specification of PGCs. However, Dpp expression begins after PGCs have budded off (and hence have been specified). It conceivable that Dpp signalling contributes to maintenance of the PGC fate, but this is not how the authors frame their observations.

2) The weakness of the phenotypes raises doubt about the importance of the mechanism invoked by the authors.

3) The absence of a clear topographical relation between the pattern of Dpp expression and the location of pole cells raises the question of whether Dpp is a physiologically relevant germ cell factor during early embryogenesis.

4) There is a lack of mechanistic insight; no molecular mechanism is proposed as to how downstream component of Dpp signalling feed into PGC fate maintenance.

5) As performed, the manipulation of Dpp signalling (admittedly difficult at this stage) lack precision in that they do not solely target germ cells or somatic cells. Specifically, RNAi againt Tkv target both pole cells and somatic cells. Definite proof that Dpp signal transduction is specifically required in pole cells would necessitate the transplantation of tkv mutant cells in an otherwise tkv wt embryo (doable but probably challenging for students educated in the modern era).

Overall, this paper makes an interesting suggestion that PGC fate specification/maintenance could be modulated by non-cell autonomous mechanism in Drosophila. However, because of logical inconsistencies and lack of mechanistic understanding, this paper fails to make a convincing case (at least so far). It may be that I missed something in the logical flow and I would be happy to be proven wrong but this would require a much improved manuscript.

Additional comment.

I am a little confused by the experiment illustrated in Fig 6, which shows that pole plasm components are delocalised when Dpp signaling is compromised. This seems inconsistent with the known timing of Dpp expression and pole cell formation. My understanding is that pole cells have formed (and therefore enclosed germplasm) before the onset of Dpp expression. How then could loss of Dpp lead to the release of pole plasm components? Again, I may have missed something but the authors do not make it easy to the reader, especially by specifying a 0-4 hrs stage in the legend (even though precise staging is relatively easy in early Drosophila embryos).

Reviewer #2: This is a beautifully crafted and conceptualized study of germ cell specification that is an important new milestone in our understanding of the germ line development. My suggestions are minor. Strong and convincing evidence is provided to show that Dpp has a critical role at syncytial blastoderm stages and mention is made that Dpp is also important at later embryo stages as well, but no explicit statement or model is offered to suggest where the Dpp sources might be. Perhaps the authors could describe their thinking? My other comments are the small details described in the specific suggestions that follow.

Abstract

Ln23 Confusing as not clear if “deployed refers to in a particular animal or, as apparently intended, in biology.

Ln 24 is a signal is a pathway

Ln 28 not clear if “longstanding model” refers to epigenesis in mammals or preformation in Drosophila

Author Summary

Ln 37 “to insulate newly formed PGCs from the adverse effects of the cell-cell signaling pathways” - is this an assumption or is it really proven to be the one and only reason for attenuating the cell cycle?

Introduction

Ln 63 what is a “somatic” gene?

Ln 65 Because all the lineages are fixed in worms, is it correct to distinguish one product of the first division as “set aside”?

Ln 69 What cycle is this minor wave of ZGA?

Ln 71 What cycle is this major wave of ZGA?

Ln 85 Not clear what the new term “model invertebrates” refers to – flies and worms? Does “higher animals” refer to mouse and human, or to other vertebrates as well?

Results

Ln 172 The described result implies that twi-Gal4 is expressed and effects expression of UAS targets at stage 5/6. It would be reassuring to examine expression of a UAS target directly, perhaps UAS-GFP?

Fig 1 Please show twiGal4 expression at the stages analyzed for Cyclin B, pH3, and PGC number.

Ln 216 Perhaps change “significantly” to “detectably”?

Ln 276 Change “embryos” to “embryo”

Ln 408 Please explain why dpERK was monitored

Ln 513 Adhesion is assumed, not demonstrated by these observations of distribution/localization

M&M

Ln 652 Michael O’Connor mis-spelled

**Have all data underlying the figures and results presented in the manuscript been provided?**

Reviewer #1: Yes

Reviewer #2: Yes

PLOS authors have the option to publish the peer review history of their article (what does this mean?). If published, this will include your full peer review and any attached files.

Reviewer #1: No

Reviewer #2: No

---

## [Decision Letter · Decision Letter 1]

9 Dec 2021

Dear Dr Deshpande,

Thank you very much for submitting your Research Article entitled 'Preformation and epigenesis converge to specify primordial germ cell fate in the early Drosophila embryo' to PLOS Genetics, and sincere apologies for the long delay in getting this decision letter to you.

The manuscript was fully evaluated at the editorial level and by independent peer reviewers. The reviewers appreciated the attention to an important topic but identified some concerns that we ask you address in a revised manuscript. There is one remaining concern from Reviewer #1 that we would like you to address. The reviewer writes: "The main problem for the first draft was that the paper did not address the functional significance of BMP signalling in germ cell specification. Unfortunately, the authors did not add experimental evidence to strengthen this statement in the current version. They attribute the weakness of the phenotype to the hypomorphic dpp and tkv alleles they used, but not explain why they did not use the stronger LOF alleles instead." Although we are not asking for any additional experiments, we do ask for you to please address this point in the discussion to specifically consider how/if use of stronger loss-of-function dpp and tky alleles might change any of your interpretations. This seems to be an important point to clarify that can only help readers of your manuscript better appreciate the impact of your work. Once this issue is addressed we will be able to render a final editorial decision without further external review.

We therefore ask you to modify the manuscript according to the review recommendations. Your revisions should address the specific points made by each reviewer.

[LINK]

Yours sincerely,

Giovanni Bosco, Ph.D.

Associate Editor

PLOS Genetics

Gregory P. Copenhaver

Editor-in-Chief

PLOS Genetics

Reviewer's Responses to Questions

**Comments to the Authors:**

Reviewer #1: The main problem for the first draft was that the paper did not address the functional significance of BMP signalling in germ cell specification. Unfortunately, the authors did not add experimental evidence to strengthen this statement in the current version. They attribute the weakness of the phenotype to the hypomorphic dpp and tkv alleles they used, but not explain why they did not use the stronger LOF alleles instead. Moreover, they claim that it is difficult to distinguish homozygous mutant embryos at the blastoderm stage, which limited them from more careful analysis. But my understanding is that this is possible with Cyo Twist-GFP or CyO,eve-lacZ. Moreover, there remain problems with imaging quality - For example, the figures presented in the response letter (on page 11) showing the distribution of dpp RNA is not informative due to extremely high background signals.

In summary, I cannot see how the current version constitutes a significant improvement over the previous one.

Reviewer #2: Very interesting work.

**Have all data underlying the figures and results presented in the manuscript been provided?**

Reviewer #1: Yes

Reviewer #2: Yes

PLOS authors have the option to publish the peer review history of their article (what does this mean?). If published, this will include your full peer review and any attached files.

Reviewer #1: No

Reviewer #2: No

---

## [Editor Report · Decision Letter 2]

17 Dec 2021

Dear Dr Deshpande,

We are pleased to inform you that your manuscript entitled "Preformation and epigenesis converge to specify primordial germ cell fate in the early Drosophila embryo" has been editorially accepted for publication in PLOS Genetics. Congratulations!

Yours sincerely,

Giovanni Bosco, Ph.D.

Associate Editor

PLOS Genetics

Gregory P. Copenhaver

Editor-in-Chief

PLOS Genetics

Comments from the reviewers (if applicable):

**Data Deposition**

http://datadryad.org/submit?journalID=pgenetics&manu=PGENETICS-D-21-01030R2

**Press Queries**

---

## [Editor Report · Acceptance letter]

30 Dec 2021

PGENETICS-D-21-01030R2 

Preformation and epigenesis converge to specify primordial germ cell fate in the early Drosophila embryo 

Dear Dr Deshpande, 

We are pleased to inform you that your manuscript entitled "Preformation and epigenesis converge to specify primordial germ cell fate in the early Drosophila embryo" has been formally accepted for publication in PLOS Genetics! Your manuscript is now with our production department and you will be notified of the publication date in due course.

With kind regards,

Livia Horvath

PLOS Genetics

On behalf of:
